# The Virulence Potential of Livestock-Associated Methicillin-Resistant *Staphylococcus aureus* Cultured from the Airways of Cystic Fibrosis Patients

**DOI:** 10.3390/toxins12060360

**Published:** 2020-05-30

**Authors:** Janina Treffon, Sarah Ann Fotiadis, Sarah van Alen, Karsten Becker, Barbara C. Kahl

**Affiliations:** 1Institute of Medical Microbiology, University Hospital Münster, 48149 Münster, Germany; Janina.Treffon@ukmuenster.de (J.T.); sarahannfotiadis@web.de (S.A.F.); sarah.vanalen@gmail.com (S.v.A.); Karsten.Becker@med.uni-greifswald.de (K.B.); 2Institute of Hygiene, University Hospital Münster, 48149 Münster, Germany; 3Business Unit Pain, Grünenthal GmbH, 52222 Stolberg, Germany; 4Friedrich Loeffler-Institute of Medical Microbiology, University Medicine Greifswald, 17475 Greifswald, Germany

**Keywords:** LA-, HA-MRSA, MSSA, cystic fibrosis, virulence, hemolysis, biofilm, invasion, cytotoxicity

## Abstract

*Staphylococcus aureus* is one of the most common pathogens that infects the airways of patients with cystic fibrosis (CF) and contributes to respiratory failure. Recently, livestock-associated methicillin-resistant *S. aureus* (LA-MRSA), usually cultured in farm animals, were detected in CF airways. Although some of these strains are able to establish severe infections in humans, there is limited knowledge about the role of LA-MRSA virulence in CF lung disease. To address this issue, we analyzed LA-MRSA, hospital-associated (HA-) MRSA and methicillin-susceptible *S. aureus* (MSSA) clinical isolates recovered early in the course of airway infection and several years after persistence in this hostile environment from pulmonary specimens of nine CF patients regarding important virulence traits such as their hemolytic activity, biofilm formation, invasion in airway epithelial cells, cytotoxicity, and antibiotic susceptibility. We detected that CF LA-MRSA isolates were resistant to tetracycline, more hemolytic and cytotoxic than HA-MRSA, and more invasive than MSSA. Despite the residence in the animal host, LA-MRSA still represent a serious threat to humans, as such clones possess a virulence potential similar or even higher than that of HA-MRSA. Furthermore, we confirmed that *S. aureus* individually adapts to the airways of CF patients, which eventually impedes the success of antistaphylococcal therapy of airway infections in CF.

## 1. Introduction

*Staphylococcus aureus* is one of the most common pathogens which infects and persists in the airways of cystic fibrosis (CF) patients [1,2]. CF is a multi-systemic, hereditary disease caused by mutations in the CF transmembrane conductance regulator gene that impair ion and water transport across epithelial membranes [3]. Consequently, patients with CF suffer from airway dehydration, mucus accumulation, and a dysregulated immune system [4], resulting in bacterial airway infections [1] that cause a decline in lung function [5] and reduced life expectancy [6]. Therefore, one of the pillars of CF therapy is the antibiotic treatment of bacterial airway infections [7]. Due to the extensive application of β-lactam antibiotics in the past, methicillin-resistant *S. aureus* (MRSA) emerged in healthcare settings and the healthy community. Currently, these so-called hospital-associated (HA-) and community-associated (CA-) MRSA represent an enormous threat, as they are highly virulent and difficult to eradicate [8,9]. Animals from livestock farms can be colonized or infected by MRSA [10]. These livestock-associated (LA-) MRSA, which most likely were derived from human methicillin-susceptible *S. aureus* (MSSA) [10], predominantly colonize animal hosts, but can be transferred back and may provoke severe diseases in humans [11,12]. LA-MRSA are a distinct subgroup of MRSA predominantly associated with the multilocus sequence typing (MLST) clonal complex (CC) 398 and harbor staphylococcal protein A gene (*spa*)-types t011, t034, t108, and close relatives. Another feature of LA-MRSA is the carriage of the resistance gene *tetM*, which confers resistance against tetracycline [10]. Numerous virulence factors enable *S. aureus* to induce and sustain long-term infections in humans and animals. Adhesins, located on the bacterial cell surface, interact with plasma proteins or extracellular matrix components of the host and facilitate bacterial adhesion [13]. Subsequently, bacterial proliferation, systemic spread, and toxinosis are provoked by hemolysins, leukocidins, exotoxins, enterotoxins, and proteases that lyse host cells, degrade tissue, and contribute to nutrient acquisition [14]. *S. aureus* possesses α-, β-, γ-, and δ-hemolysins that are under the control of the accessory gene regulator (*agr*) and predominantly lyse erythrocytes by either forming pores in the host cell membranes or degrading cell wall components [15,16]. In addition, *S. aureus* is able to hide from the host immune system by expressing immunomodulatory factors, like *chps* (chemotaxis inhibitory protein of *S. aureus*) and *scin* (staphylococcal complement inhibitor), located on bacteriophages [17]. Additionally, the formation of biofilms, which are multicellular agglomerations embedded in a matrix formed by proteins, polysaccharides, and/or extracellular DNA [18], supports bacterial immune evasion and presents an adaptation advantage during staphylococcal persistence in CF airways [19]. Furthermore, the bacterial invasion of host cells, mediated by a number of bacterial adhesion proteins, provides additional protection from the host immune system and antibiotic therapy, and is suggested to contribute to recurrent staphylococcal infections and long-term persistence in CF patients [20,21].

Respiratory infections caused by MRSA are associated with more severe lung disease compared to MSSA in CF patients [6]. Quite recently, LA-MRSA were detected in the airways of CF patients [22,23,24]. However, although some studies analyzed the virulence and pathogenicity of LA-MRSA recently [25], there is only limited knowledge about the role of LA-MRSA virulence in CF lung disease, especially if such LA-MRSA clones persist in the airways for extended periods. Therefore, we characterized 18 LA-MRSA, HA-MRSA, and MSSA clinical isolates of nine CF patients, which were isolated early in the course of airway infection and several years after persistence in this hostile niche. These isolates were analyzed by MLST and *spa*-genotyping and further characterized regarding their hemolytic activity, biofilm formation, invasion in airway epithelial cells, cytotoxicity, and antibiotic resistance. We detected that CF LA-MRSA isolates were more hemolytic and cytotoxic than HA-MRSA and more invasive than MSSA, indicating that despite their prior residence in the animal host, they still have a highly virulent phenotype in humans.

## 2. Results

### 2.1. LA-MRSA Are Strongly Hemolytic

The hemolysins of *S. aureus* differ in their affinity to red blood cells of various species. While α-hemolysin has a high affinity for rabbit erythrocytes, which express high amounts of the α-hemolysin receptor ADAM-10 (a disintegrin and metalloprotease 10) on their surface [16], the activity of the sphingomyelinase β-hemolysin is more dedicated to the sphingomyelin-rich membrane of sheep erythrocytes [26]. To get a broad overview of the hemolytic activity of LA-MRSA, HA-MRSA, and MSSA clinical CF isolates, we (a) plated bacterial strains on Columbia sheep blood agar and assessed hemolysis macroscopically after incubation at 37 °C, and subsequently 4 °C, for 24 h, and (b) incubated rabbit erythrocytes with bacterial supernatants, measured the absorbance of released hemoglobin, and determined the dilution step at which 50% of all erythrocytes were lysed (ED_50_). In addition, a microarray analysis was performed to detect hemolysis-associated virulence genes. After incubation at 37 °C on sheep blood agar, all LA-MRSA and most HA-MRSA and MSSA revealed narrow to broad zones of clear hemolysis (Table 1, Appendix A), indicative of α-hemolysin production. However, only the hemolytic halo of the LA-MRSA strains and the late HA-MRSA isolate t548 was surrounded by a large dark, diffuse zone of hemolysis, which became clear after overnight incubation at 4 °C (Table 1, Appendix A). This “hot–cold hemolysis” is indicative of β-hemolysin production [16]. All isolates that were strongly hemolytic on sheep blood agar had a high ED_50_ and thus a strong hemolytic effect on rabbit erythrocytes in the photometrical quantification assay, except for the late MSSA isolate t002 (Figure 1a). Interestingly, two late HA-MRSA isolates (t003 and t548) were significantly more hemolytic compared to their corresponding early isolates. Grouping the *S. aureus* isolates according to their *agr*-type revealed that *agrI* is the most hemolytic type, followed by *agrII* and *agrIII* (Figure 1b). Interestingly, while all LA-MRSA and HA-MRSA could be assigned to *agrI* and II, respectively, MSSA belonged to *agr*-type I, II, and III. Furthermore, the microarray analysis revealed that all bacterial strains harbor the genes *hla*, *hld*, and *hlgA* encoding for α-, δ- and γ-hemolysins, but only the LA-MRSA isolates and the late HA-MRSA isolate t548 carried the intact *hlb* encoding for β-hemolysin (Figure 1c). Importantly, the absence of the intact *hlb* was accompanied by the presence of *sak* (staphylokinase), *chps*, and *scin* (Appendix A), indicating that *hlb* was interrupted by the integration of a bacteriophage that carries these three immune evasion genes.

### 2.2. LA-MRSA Are Weak Biofilm Formers

To assess the biofilm-forming capacity of LA-MRSA, HA-MRSA, and MSSA clinical CF isolates, we raised bacterial biofilms in microtiter plates and stained them with crystal violet. In addition, a microarray analysis was done to detect biofilm-associated genes. Only the late isolates of LA-MRSA t034T and MSSA t080 formed biofilms when incubated in microtiter plates (Figure 2a), although the microarray analysis revealed the presence of *icaA*, *C*, and *D*, which encode for the most common staphylococcal biofilm component, polysaccharide intercellular adhesin PIA [16], in all clinical isolates (Figure 2b).

### 2.3. LA-MRSA Invade Airway Epithelial Cells and Are Cytotoxic

To determine the invasiveness and cytotoxicity of LA-MRSA, HA-MRSA, and MSSA clinical CF isolates towards host cells, we infected A549 airways epithelial cells and measured the number of intracellularly located bacteria 2 h post infection and the release of lactate dehydrogenase (LDH) from A549 cells, which is released upon necrosis [27], 24 h post infection. Only the late isolates of LA-MRSA t034T and HA-MRSA t010 were significantly more invasive in A549 cells than their respective early isolates and the highly invasive control strain Cowan I (Figure 3a). All other isolates were either equally or less invasive compared to the control strain. Nevertheless, of all the analyzed isolates, the MSSA strains demonstrated the lowest invasiveness. Surprisingly, this group harbored the highest number of cytotoxic strains (Figure 4a). In more detail, the cytotoxicity assay revealed that four MSSA strains (*spa*-types t617 and t080), three LA-MRSA strains (early and late LA-MRSA t034S and late LA-MRSA t034T), but only one HA-MRSA strain (late HA-MRSA t003), provoked LDH release from A549 cells 24 h post infection.

By microarray analysis, we detected a similar pattern of adhesion-, invasion-, and cytotoxicity-associated genes in all isolates, independent of their ability to invade or lyse host cells (Figure 3b and Figure 4b). However, in contrast to all HA-MRSA and some MSSA strains, all LA-MRSA isolates harbored the collagen adhesin gene *cna*, but were negative for *fib* and *sasG* encoding for fibrinogen-binding protein and surface protein G (Figure 3b) and the cytotoxicity-associated enterotoxin, leukocidin, exotoxin, and serine protease genes *egc*, *lukD/E*, *seg/i/lm/ln/lo/lu*, *setB2/B3*, and *splA/B* (Figure 4b).

### 2.4. LA-MRSA Are Tetracycline-Resistant

The presence of specific resistance genes in LA-MRSA, HA-MRSA, and MSSA clinical CF isolates was determined by microarray analysis. As expected, LA- and HA-MRSA carried more resistance genes than MSSA (Table 2). Nearly all analyzed bacterial strains harbored the *bla*-operon conferring penicillin resistance, but only LA- and HA-MRSA were positive for the β-lactam resistance gene *mecA*. In addition, most LA- and HA-MRSA isolates, but only two MSSA strains, possessed the erythromycin and clindamycin resistance gene *ermA*. Interestingly, while *fosB* (fosfomycin resistance gene) was detectable in all HA-MRSA and most MSSA, *tetM* and *tetK* (tetracycline resistance genes) were exclusively found in LA-MRSA. Importantly, in the late isolate of MSSA t080, the genes *aacA-aphD* (gentamicin, tobramycin, and kanamycin resistance) and *dfrS1* (trimethoprim resistance) were not present, although we identified them in the corresponding early isolate, indicating a loss of the resistance genes in the course of adaptation to the host. Additionally, in the late isolate of LA-MRSA t034T and LA-MRSA t011, the genes for *aadD* (kanamycin resistance) and *ermC* (erythromycin resistance) were lost during persistence, respectively. In contrast, the genes *aphA3*, *cat*, *cfr*, *fexA*, *inuA*, *mefA*, *mphC*, *msrA*, *mupA*, *vanA/B/Z*, *vatA/B*, and *vgaA* conferring resistance to aminoglycosides, phenicol antibiotics, lincosamides, macrolides, mupirocin, glycopeptides, and streptogramins, respectively, were not detected at all in any of the analyzed bacterial strains.

## 3. Discussion

To the best of our knowledge, this is the first study that analyzed the virulence and adaptation of LA-MRSA infecting the respiratory tract of CF patients. The study was performed in the Münster region, an area with a high pig-farming density and known as a hotspot of LA-MRSA in Germany [28,29]. LA-MRSA CF isolates were examined in comparison to HA-MRSA and MSSA CF isolates regarding their hemolytic activity, biofilm formation, invasion in airway epithelial cells, cytotoxicity, and resistance genes.

Our hemolysis assay revealed that LA-MRSA were significantly more hemolytic than HA-MRSA and MSSA, which might be partly due to the fact that these clones belong to the *agr*-specificity group I. This is in accordance with the findings of Mairpady Shambat et al., who showed that *S. aureus* isolates of *agr*-type I had a higher hemolytic activity than *S. aureus* assigned to *agrII* and *III* [30]. Importantly, all LA-MRSA and only one HA-MRSA strain produced β-hemolysin. As these isolates were negative for *sak*, *chps*, and *scin*, genes that are encoded on a bacteriophage [17], it is highly likely that they lost this phage in the course of bacterial adaptation to the animal host, in which genes which provide protection against the human host defense, are not required and are dispensable [10], while a loss during chronic airway infection of the respective CF patient is less likely in these LA-MRSA isolates [31]. However, as for other human patient populations, bacteriophage phi3 acquisition seemed to be no major driver for the re-adaptation of LA-MRSA to the human host [32].

All LA-MRSA in our study could be assigned to CC398 and *spa*-types t011 and t034, which are the major LA-MRSA-associated *spa*-types recovered from farm animals, companion animals, and humans in Germany and Europe in general [33,34,35]. Interestingly, two other research groups isolated LA-MRSA of the same clonal complex and *spa*-types from CF airways, which also did not carry the bacteriophage-associated genes *sak*, *chps*, and *scin* either [22,24]. In our study, we also analyzed persisting LA-MRSA isolates, which resided in the airways of the individual CF patients for three, four, and eight years. Unexpectedly, these LA-MRSA isolates did not pick up the bacteriophage, which confers protection against host defense, and which could have been potentially transferred by co-colonizing *S. aureus* strains.

Similar to HA-MRSA and MSSA, LA-MRSA were weak biofilm formers. Therefore, we assume that biofilm production is not associated with the sole presence of specific biofilm- or adhesion-associated genes like *icaADC*, *clfA/B* (clumping factor A/B), *fnbA/B* (fibronectin-binding protein A/B), *sasG*, and *sdrC/D* (serine-aspartate repeat-containing protein C/D) [19,36]. Different regulatory mechanisms might induce or repress biofilm formation in the investigated bacterial isolates. Furthermore, the biofilm production of MRSA is heavily influenced by environmental factors, which strongly differ between in vitro and in vivo experiments and might cause the conversion of in vivo biofilm formers to in vitro non-biofilm formers [37]. Thus, in vivo biofilm experiments should be done to investigate the biofilm production of the *S. aureus* CF isolates in more detail.

A previous study with porcine and non-CF human LA-MRSA revealed a high cytotoxic potential comparable to certain CA-MRSA [38]. Here, in interaction studies with airway epithelial cells, LA- and HA-MRSA tended to be more invasive than MSSA. Surprisingly, LA-MRSA and MSSA strains were especially cytotoxic and caused LDH release from airway epithelial cells, indicating that not only high, but also low, numbers of intracellular bacteria can induce host cell necrosis. Interestingly, while the adhesion- and invasion-associated gene cluster was similar among LA-MRSA and MSSA, only LA-MRSA lacked toxin and protease genes important for infections in humans. However, invasive and cytotoxic bacteria harbored the same adhesion-, invasion-, and cytotoxicity-associated genes as non-invasive and non-cytotoxic strains, indicating that these genes can, but not necessarily must, support the bacterial invasion and lysis of host cells. To further study these effects, gene expression analysis will be helpful to determine if differences in transcription might cause the respective phenotype. In addition, the investigation of virulence regulators such as sigma factor B (*sigB*), staphylococcal accessory regulator A (*sarA*), and *agr* or genes of the bacterial stress response, like superoxide dismutases *sodA* and *sodM*, or the caseinolytic protease *clpC*, which were shown to be involved in intracellular survival and long-term persistence of *S. aureus* in non-professional phagocytes [39,40,41,42], could be helpful to explore the different invasion and cytotoxicity traits of the bacterial CF isolates. 

As expected, the microarray analysis verified the presence of the methicillin-resistance gene *mecA* in LA- and HA-MRSA [8]. Interestingly, *tetM* and *tetK* conferring resistance to tetracycline were exclusively found in LA-MRSA, which most likely represents bacterial adaptation to the specific antibiotic selective pressure in livestock animals [10] and might result from the addition of tetracycline to feedstuff with the aim of preventing and controlling infections in livestock [43]. Recently, a Brazilian and a Belgian group also isolated tetracycline-resistant LA-MRSA from the airways of CF patients [22,24]. However, resistance to this antibiotic is not essentially part of the LA-MRSA resistance profile, as Garbacz et al. found an LA-MRSA CF isolate that was tetracycline-susceptible [23]. The presence of all remaining antibiotic resistance genes mostly detected in MRSA, but sporadically also found in MSSA (*aacA*-*aphD*, *aadD*, *blaI/R/Z*, *dfrS1*, *ermA/B/C*, and *fosB*), most likely results from the application of tobramycin, kanamycin, penicillins, trimethoprim, erythromycin, and fosfomycin to CF patients or livestock animals. However, as the analyzed *S. aureus* strains are still sensitive to a lot of antibiotics commonly used in CF therapy or livestock infection prevention, like vancomycin, linezolid, chloramphenicol, clindamycin, lincomycin, mupirocin, streptogramins, and several macrolides [43,44], many therapeutic options to cure infections caused by these bacterial strains are still available.

In addition to the direct comparison between LA-MRSA, HA-MRSA, and MSSA, we investigated differences between early and late *S. aureus* CF isolates and studied the bacterial adaptation mechanism facilitating long-term persistence in CF airways. We detected differences in the phenotypic characteristics of late, compared to early, isolates: Of the nine investigated early/late strain pairs, some late isolates were significantly more hemolytic (HA-MRSA t003 and t548) or less hemolytic (MSSA t002), more cytotoxic (LA-MRSA t034T), more invasive (LA-MRSA t034T and HA-MRSA t010), produced more biofilm (LA-MRSA t034T and MSSA t080), and possessed fewer resistance genes (LA-MRSA t034T and t011 and MSSA t080) than their respective early counterparts. These findings reveal that changes in hemolysis, invasion, cytotoxicity, biofilm formation, and resistances can be associated with bacterial long-term persistence in CF airways, as also detected elsewhere [2,19,21,45]. Importantly, these adaptations were only detectable in a few isolates, indicating that *S. aureus* adaptation to CF airways is an individual, complex process that probably depends on various factors including host, treatment, co-infecting isolates, and the state of lung disease.

Summarized, we demonstrated that LA-MRSA infecting CF airways are strongly hemolytic, cytotoxic, and invasive bacteria that possess a virulence potential similar or even higher than that of HA-MRSA. Furthermore, we confirmed that *S. aureus* individually adapts to the airways of CF patients, which eventually could impede the success of antistaphylococcal therapy of respiratory CF infections.

## 4. Materials and Methods

### 4.1. Ethical Statement

An ethical approvement was obtained from the Ethical Committee in Münster, Westfalian Wilhelms University and Physicians Chamber Westfalen-Lippe, Germany (2018-466-f-S), approval date: 20 February 2019 Due to the retrospective study and anonymized patient data, no written informed consent was required. 

### 4.2. Bacterial Strains and Growth Conditions

Clinical *S. aureus* strains were isolated from the sputum samples or throat swabs of nine chronically *S. aureus*-infected CF patients from the CF centers of the University Hospital Münster, Germany, and the Clemenshospital, Münster, Germany, and were processed as described recently [40]. Per patient, two clonal *S. aureus* isolates, an early and a late isolate, which carried the same *spa*-type and MLST, were selected for analysis. While the early isolates were obtained soon after the onset of staphylococcal airway infection, the late isolates were recovered three to 13 years later and represent persistent bacterial strains. Details about the investigated clinical *S. aureus* CF isolates are shown in Table 3. The laboratory strains *S. aureus* Cowan I, *Staphylococcus epidermidis* RP62A, and *Staphylococcus carnosus* TM300 were used as controls and are listed in Table 4.

In general, bacterial glycerol stocks were subcultured on Columbia sheep blood agar (37 °C, 18–24 h) and streaked onto fresh blood agar plates several days prior to the experiments. For the analysis of hemolysis, bacteria were cultivated overnight for 18 h at 37 °C and 160 rpm in Tryptic Soy Broth (TSB) under aerated conditions, centrifuged at 45,000 rpm for 5 min and sterile filtered using a 0.22 µm filter. For the determination of biofilm formation, bacteria were grown in brain heart infusion supplemented with 0.25% glucose (BHIG) at 37 °C at 160 rpm under aerated conditions. For the infection of airway epithelial cells with the subsequent determination of bacterial invasion and cytotoxicity, glycerol stocks were used, generated as follows: starting from a bacterial overnight culture in TSB, 10 mL of TSB in a 100 mL baffled flask were set to an OD_578 nm_ of 0.1 and cultivated at 37 °C and 160 rpm until mid-logarithmic growth phase. Using this culture, 5 mL of phosphate-buffered saline (PBS) were inoculated to an OD_578_ nm of 1 and centrifuged at 4500 rpm for 7 min. Subsequently, the bacterial pellet was resuspended in RPMI-1640 (R7388, Sigma-Aldrich [now Merck kGaA, Darmstadt, Germany]) medium supplemented with 2 mg/mL NaHCO_3_ (Sigma-Aldrich [now Merck kGaA, Darmstadt, Germany]) and treated with a sonifier in order to separate cell clumps. Afterwards, glycerol was added to generate a 20% working solution, serially diluted in PBS, plated on blood agar, and incubated over night at 37 °C to determine colony forming units (CFU) the next day. Glycerol stocks were stored at −20 °C until usage.

### 4.3. spa-Typing

Staphylococcal DNA was isolated with the DNeasy Blood & Tissue Kit (Qiagen, Hilden, Germany), following the instructions of the manufacturer, with a lysostaphin (AMBI Products, Lawrence, NY, USA) step implemented at the beginning. The polymorphic region x of *spa* was amplified with the primers *spa*-forward (5′-TAAAGACGATCCTTCGGTGAGC-3′) and *spa*-reverse (5′-CAGCAGTAGTGCCGTTTGCTT-3′) and the REDTaq^®^ ReadyMix™ (Sigma-Aldrich [now Merck kGaA, Darmstadt, Germany]) via polymerase chain reaction (PCR), applying the following program on a thermocycler (Bio-Rad Laboratories GmbH, Feldkirchen, Germany): 1 min 95 °C, 30 s 95 °C + 30 s 60 °C + 30 s 72 °C (35 repetitions), 3 min 72 °C. The PCR product was purified using the QIAquick PCR Purification Kit (Qiagen, Hilden, Germany) and sequenced by Eurofins Genomics (Ebersberg, Germany). The *spa*-types were identified using the software Ridom^®^ StaphType (Ridom GmbH, Münster, Germany). The *spa*-type repeats are listed in Table 3.

### 4.4. Microarray Analysis

Staphylococcal DNA was isolated as described for *spa*-typing. However, instead of lysostaphin, components A1 and A2 of the *S. aureus* Genotyping Kit 2.0 (Alere Technologies GmbH [now Abbott Rapid Diagnostics GmbH, Jena, Germany]) were applied to lyse bacteria. Subsequently, microarrays were prepared according to the instructions of the *S. aureus* Genotyping Kit 2.0 and analyzed using the ArrayMate™ Reader (Alere Technologies GmbH [now Abbott Rapid Diagnostics GmbH, Jena, Germany]), running with the following program: 5 min 96 °C, 60 s 96 °C + 20 s 50 °C + 40 s 72 °C (55 repetitions). The results of all analyzed genes are listed in Appendix A.

### 4.5. Hemolysis Assay

#### 4.5.1. Macroscopic Evaluation

Bacteria were streaked onto Columbia sheep blood agar, incubated at 37 °C, and subsequently 4 °C, for 24 h, to determine the activity of bacterial hemolysins. Pictures were taken with a digital camera and cropped using IrfanView (Irfan Skiljan, Wiener Neustadt, Austria). Colony sizes of the isolates might differ due to variable camera positions and therefore, cannot be compared among each other. As determination of hemolysis, but not colony size, was focus of the investigation, colony size was not measured. 

#### 4.5.2. Photometrical Quantification

Photometrical quantification of bacterial hemolytic activity was assessed by incubating rabbit erythrocytes with sterile filtered supernatants of bacteria grown overnight in TSB and measuring the absorbance of hemoglobin released from damaged erythrocytes. Briefly, 100 µL of sterile filtered supernatants of bacterial overnight cultures in TSB were added to the wells of a 96-well plate (u-bottom) and diluted in a 1% bovine serum albumin (BSA)/PBS solution in 1:2 steps. Subsequently, 100 µL of a 2% rabbit erythrocyte solution generated from rabbit whole blood (Fiebig Nährstofftechnik, Idstein, Germany), which was three times centrifuged at 3000 rpm for 10 min at 4 °C, resuspended in 1% BSA/PBS solution, and finally diluted 1:50 in 1% BSA/PBS solution, were added to each well and mixed gently by pipetting up and down. The plate was incubated for 1 h at 37 °C to allow the lysis of erythrocytes and centrifuged for 10 min at 2000 rpm at room temperature. One hundred microliters of the supernatant were transferred into a new 96-well plate and the absorbance of hemoglobin released from damaged erythrocytes was measured at OD_450_ nm with a microplate reader (Bio-Rad Laboratories GmbH, Feldkirchen, Germany). As a negative control, untreated 2% erythrocyte solution, and as a positive control, 2% erythrocyte solution treated for 1 h with 4.5% Triton X-100, was used. To compare the hemolytic effect of the different *S. aureus* strains, the absorbance was plotted against the dilution steps (step 0 = undiluted, step 1 = two-fold dilution, step 2 = four-fold dilution, step 3 = eight-fold dilution, step 4 = 16-fold dilution, step 5 = 32-fold dilution, step 6 = 64-fold dilution, step 7 = 128-fold dilution, step 8 = 256-fold dilution, step 9 = 512-fold dilution) and the mean effective dilution ED_50_ (the dilution step at which 50% of all erythrocytes are lysed) was calculated using the linear slope of the inflection point of the hemolysis graph. The higher the ED_50_, the stronger the hemolytic effect.

### 4.6. Biofilm Assay

Bacterial biofilm formation was investigated as described by Schwartbeck et al. [19], with some modifications. Briefly, bacterial overnight cultures in BHIG were used to inoculate 2 mL of fresh BHIG to an OD_578_ nm of 0.02. Of this solution, 200 µL were added to the wells of a 96-well plate (Greiner Bio One, Frickenhausen, Germany) and incubated for 24 h at 37 °C in a metal box lined with wet paper towels to prevent dehydration. Subsequently, the plates were washed three times with 200 µL PBS/well and the biofilms were stained with 0.1% crystal violet for 15 min at room temperature. After three further washing steps, 100 µL ethanol/acetone (80:20) were added to each well to solubilize the biofilms and the absorbance was detected at 655 nm with a microplate reader (Bio-Rad Laboratories GmbH, Feldkirchen, Germany). As a negative control, *S. carnosus* TM300, and as a positive control, *S. epidermidis* RP62A, were used.

### 4.7. Cell Culture Infection Model

We performed infection assays using A549 airway epithelial cells (ATCC^®^ CCL-185™). The cells were cultivated and infected as described recently [40], with some modifications. Briefly, the cells were seeded in 75 cm^2^ cell culture flasks (Greiner Bio-One, Frickenhausen, Germany) and cultivated in RPMI-1640 (R7388, Sigma-Aldrich [now Merck kGaA, Darmstadt, Germany]) medium, supplemented with 2 mg/mL NaHCO_3_ (Sigma-Aldrich [now Merck kGaA, Darmstadt, Germany]) and 10% fetal bovine serum (FBS) (Biochrom GmbH [now Merck kGaA, Darmstadt, Germany]) at 37 °C and 5% CO_2_. The medium was changed at least every third day and the cells were split depending on their density. The detachment of confluent cells was achieved by 15 min of incubation with Trypsin/EDTA (PAA Laboratories [now Fisher Scientific GmbH, Schwerte, Germany]).

For the invasion assay, three days prior to the experiments, A549 cells were seeded in 12-well plates (Corning GmbH, Wiesbaden, Germany) at a density of 40,000 cells/cm^2^ and cultivated to confluence. For each experiment, one additional well was covered with cells, which was used to determine the cell number on the day of the experiment, when the cells were washed with PBS (D8537, Sigma-Aldrich [now Merck kGaA, Darmstadt, Germany]) and covered with invasion medium, consisting of RPMI supplemented with 2 mg/mL NaHCO_3_ and 10 mg/mL human serum albumin (HSA) (Kedrion Biopharma, Gräfelfing, Germany). Subsequently, the host cells were infected with mid-logarithmic *S. aureus* from glycerol stocks, using a multiplicity of infection (MOI) of 100, and incubated at room temperature for 15 min to allow the sedimentation of bacteria. Eukaryotic and prokaryotic cells were incubated at 37 °C and 5% CO_2_. To determine the inoculum used for infection, the bacterial glycerol stocks were serial diluted in PBS, plated on blood agar, and incubated overnight at 37 °C to determine the CFU the next day. After 2 h of incubation, cells were washed two times with PBS and extracellular bacteria were killed by the incubation of cells for 30 min with RPMI medium supplemented with 2 mg/mL NaHCO_3_ and 20 µg/mL lysostaphin. Supernatants of the killing step were plated on agar to control the toxic effect of lysostaphin. After applying two further washing steps, cells were lysed in water to release the intracellular bacteria. Cell lysates were pipetted through a syringe, serially diluted in PBS, and plated on blood agar to determine the CFU/mL of intracellular bacteria after 24 h of incubation at 37 °C. As control, the highly invasive *S. aureus* strain Cowan I was used.

### 4.8. Cytotoxicity Assay

To analyze the cytotoxicity of intracellular bacteria, A549 cells were infected and treated as described for the invasion assay. However, after lysostaphin treatment, cells were washed and covered with RPMI medium supplemented with 2 mg/mL NaHCO_3_, 10% FBS, 1% antibiotic/antimycotic solution, and 2% MycoKill AB and incubated for up to 24 h at 37 °C and 5% CO_2_. Subsequently, 500 µL of culture supernatant were drawn from the well plate, centrifuged at 5000 rpm for 5 min and used to determine the bacterial cytotoxicity by measuring the activity of LDH with the CytoTox^®^ non-radioactive cytotoxicity assay kit (Promega, Walldorf, Germany), following the manufacturer’s instructions. The higher the absorbance, the higher the LDH release from A549 cells and the bacterial cytotoxicity. As a negative control, the supernatant of A549 cells treated with 10 µL RPMI medium supplemented with 20% glycerin, and as a positive control, the supernatant of A549 cells treated with 0.09% Triton X-100, were used (the latter one applied in a 1:10 dilution).

### 4.9. Statistical Analysis

Statistical analysis was done by applying one-way analysis of variance (ANOVA) with subsequent Bonferroni’s post hoc tests for multiple comparisons (*p*-value * ≤ 0.05, *p*-value ** ≤ 0.01, *p*-value *** ≤ 0.001) using the software GraphPad Prism 5 (GraphPad Software, San Diego, CA, USA). Only the most important significant differences are indicated in the figures.

## Figures and Tables

**Figure 1 toxins-12-00360-f001:**
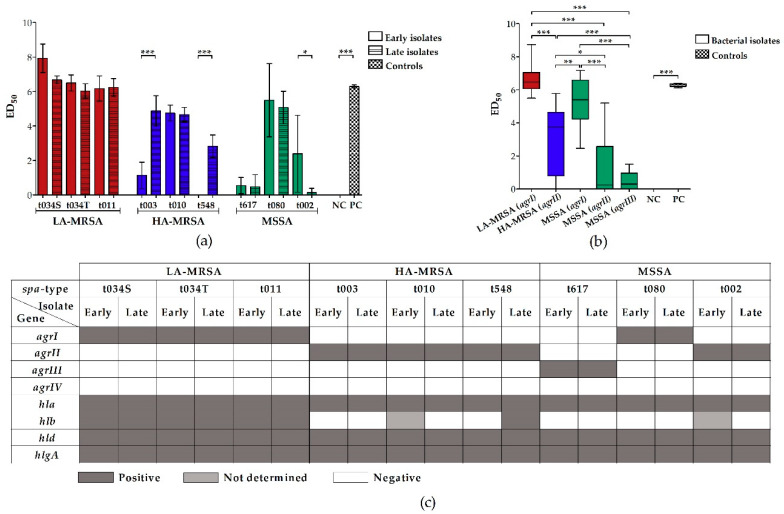
Analysis of the hemolytic effect of supernatants from clinical LA-MRSA, HA-MRSA, and MSSA CF isolates on rabbit erythrocytes, according to the *spa*-type and the *agr*-type and microarray analysis of hemolysis-associated virulence genes. (a + b) Strength of bacterial hemolytic activity was determined by incubating rabbit erythrocytes with sterile-filtered supernatants from bacterial overnight cultures and measuring the absorbance of released hemoglobin from damaged erythrocytes at OD_450 nm_. Absorbance was plotted against the dilution step. ED_50_ (the dilution step at which 50% of all erythrocytes are lysed) was calculated using the linear slope of the inflection point of the hemolysis graph. The higher the ED_50_, the stronger the hemolytic effect. As a negative control, untreated erythrocytes, and as a positive control, erythrocytes treated with Triton X-100, were used. Results were grouped according to the *spa*-type (**a**) and the *agr*-type (**b**) of the clinical isolates. All LA-MRSA belonged to *agr*-type I, all HA-MRSA to *agr*-type II. MSSA could be divided into *agr*-type I (*spa*-type t080), *agr*-type II (*spa*-type t002), and *agr*-type III (*spa*-type t617). (**a**) Data represent the mean of at least four biological replicates ± SD. (**b**) Biological replicates of the strains grouped according to their *agr*-type were averaged and are presented ± SD. Negative values (no hemolysis, no dilution required) were set to zero. Statistical analysis was done by applying one-way ANOVA, followed by Bonferroni’s post hoc test for multiple comparisons (* *p* ≤ 0.05, ** *p* ≤ 0.01, *** *p* ≤ 0.001). To keep the graph clear, only particular significances are displayed. (**c**) Microarray was performed according to the manufacturer’s instructions (Alere Technologies GmbH [now Abbott Rapid Diagnostics GmbH, Jena, Germany]). While dark gray squares indicate genes that were detected, white squares mark negative results. Light gray squares represent genes whose presence was not determined. Abbreviations: *agrI/II/III/IV* (accessory gene regulator I/II/III/IV), ANOVA (analysis of variance), ED_50_ (mean effective dilution), HA-MRSA (hospital-associated methicillin-resistant *S. aureus*), *hla/b/d* (hemolysin α/β/δ), *hlgA* (hemolysin γ component A), LA-MRSA (livestock-associated methicillin-resistant *S. aureus*), MSSA (methicillin-susceptible *S. aureus*), NC (negative control), PC (positive control), S (isolated from sputum), SD (standard deviation), *spa* (staphylococcal protein A), T (isolated from throat).

**Figure 2 toxins-12-00360-f002:**
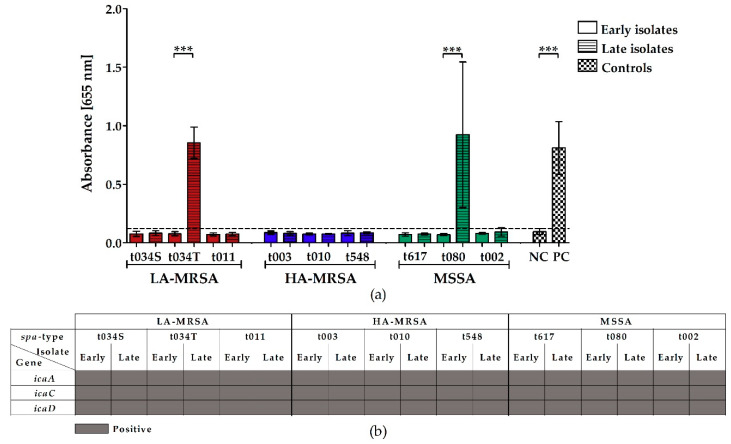
Biofilm formation of clinical LA-MRSA, HA-MRSA, and MSSA CF isolates in BHIG and microarray analysis of biofilm-associated virulence genes. (**a**) Bacterial overnight cultures were diluted in fresh BHIG, transferred into a 96-well plate and incubated at 37 °C to allow biofilm formation. After 24 h, the biofilms were washed, stained with crystal violet, solubilized with ethanol/acetone, and absorbance was measured at OD_655_ nm with a microplate reader. As a negative control, *Staphylococcus carnosus* TM300, and as a positive control, *Staphylococcus epidermidis* RP62A, were used. Data represent the mean of at least five biological replicates ± SD, analyzed in four to eight technical repetitions. Statistical analysis was done by applying one-way ANOVA, followed by Bonferroni’s post hoc test for multiple comparisons (*** *p* ≤ 0.001). To keep the graph clear, only particular significances are displayed. (**b**) Microarray was performed according to the manufacturer’s instructions (Alere Technologies GmbH [now Abbott Rapid Diagnostics GmbH, Jena, Germany]). Dark gray squares indicate genes that were detected. Abbreviations: ANOVA (analysis of variance), BHIG (brain heart infusion supplemented with 0.25% glucose), HA-MRSA (hospital-associated methicillin-resistant *S. aureus*), *icaA/C/D* (intercellular adhesion protein A/C/D), LA-MRSA (livestock-associated methicillin-resistant *S. aureus*), MSSA (methicillin-susceptible *S. aureus*), NC (negative control), PC (positive control), S (isolated from sputum), SD (standard deviation), *spa* (staphylococcal protein A), T (isolated from throat).

**Figure 3 toxins-12-00360-f003:**
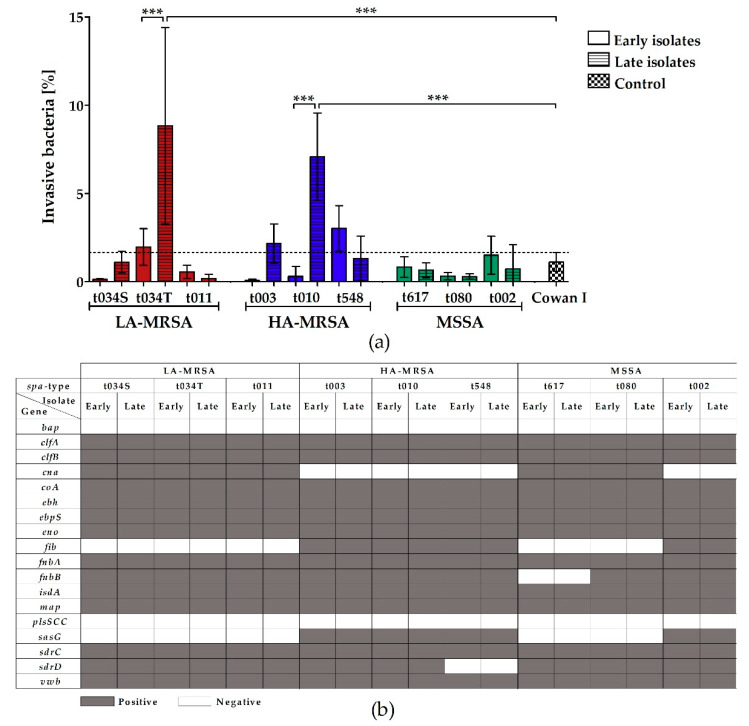
Invasion of clinical LA-MRSA, HA-MRSA, and MSSA CF isolates into A549 airway epithelial cells and microarray analysis of adhesion and invasion-associated virulence genes. (**a**) Epithelial lung cells were infected with mid-logarithmic *S. aureus* from glycerol stocks and incubated for 2 h, followed by the eradication of extracellular bacteria. Cells were lysed in water and intracellular bacteria were plated on agar to determine the number of internalized bacteria by counting CFU. The percentage of invasive bacteria was calculated by dividing CFU/mL of internalized bacteria by CFU/mL of the inoculum used for infection. As a control, the highly invasive *S. aureus* strain Cowan I was used. Data represent the mean of at least four biological replicates ± SD, analyzed in technical duplicates. Statistical analysis was done by applying one-way ANOVA, followed by Bonferroni’s post hoc test for multiple comparisons (*** *p* ≤ 0.001). To keep the graph clear, only particular significances are displayed. (**b**) Microarray was performed according to the manufacturer’s instructions (Alere Technologies GmbH [now Abbott Rapid Diagnostics GmbH, Jena, Germany]). While dark gray squares indicate the genes that were detected, white squares mark undetected genes. Abbreviations: ANOVA (analysis of variance), *bap* (biofilm-associated protein), CFU (colony forming units), *clfA/B* (clumping factor A/B), *cna* (collagen adhesin), *coA* (staphylocoagulase), *ebh* (extracellular matrix-binding protein), *ebpS* (elastin-binding protein), *eno* (enolase), *fib* (fibrinogen-binding protein), *fnbA/B* (fibronectin-binding protein A/B), HA-MRSA (hospital-associated methicillin-resistant *S. aureus*), *isdA* (iron-regulated surface determinant protein), LA-MRSA (livestock-associated methicillin-resistant *S. aureus*), *map* (MHC class II analog protein), MSSA (methicillin-susceptible *S. aureus*), *plsSCC* (plasmin-sensitive surface protein encoded on staphylococcal cassette chromosome), S (isolated from sputum), *sasG* (surface protein G), SD (standard deviation), *sdrC/D* (serine-aspartate repeat-containing protein C/D), *spa* (staphylococcal protein A), T (isolated from throat), *vwb* (von Willebrand factor-binding protein).

**Figure 4 toxins-12-00360-f004:**
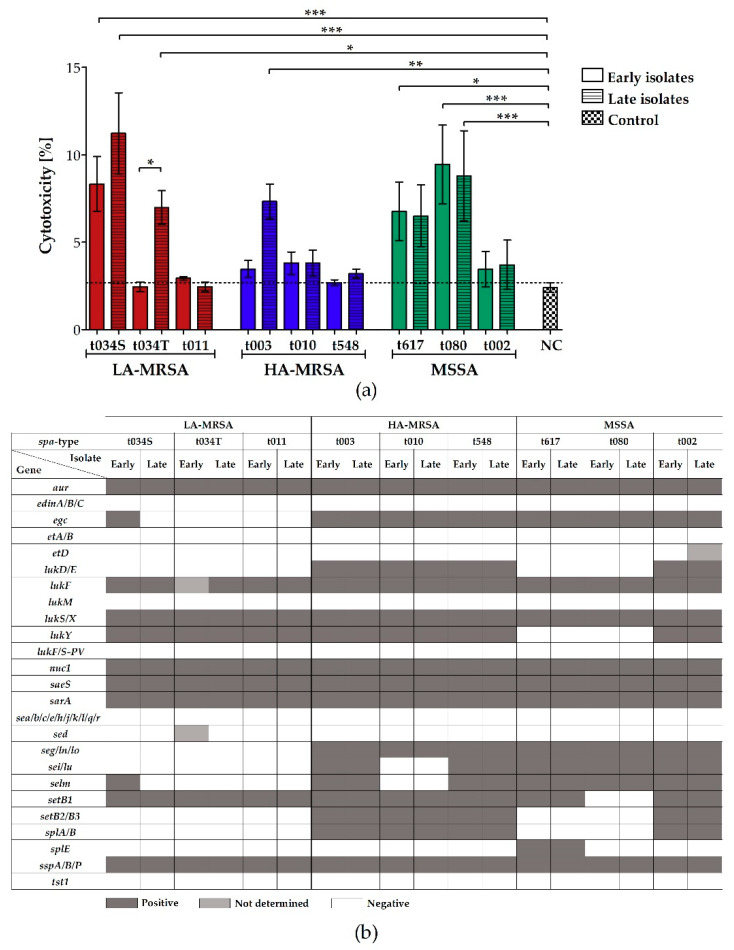
Cytotoxic effect of clinical LA-MRSA, HA-MRSA, and MSSA CF isolates on A549 airway epithelial cells and microarray analysis of cytotoxicity-associated virulence genes. (**a**) The cytotoxic effect of intracellular bacteria was assessed by measuring the amount of released LDH in the culture supernatant of A549 cells 24 h post infection following the instructions of the CytoTox^®^ non-radioactive cytotoxicity assay kit (Promega, Walldorf, Germany). As a negative control, the supernatant of A549 cells treated with RPMI medium supplemented with 20% glycerin, and as a positive control, supernatant of A549 cells treated with 0.09% Triton X-100, were used (the latter one applied in 1:10 dilution). The percentage of the cytotoxicity of the bacterial strains and negative control was calculated with regard to the positive control, which was set to 100%. Data represent the mean of three biological replicates ± SD, analyzed in technical duplicates. Statistical analysis was done by applying one-way ANOVA, followed by Bonferroni’s post hoc test for multiple comparisons (* *p* ≤ 0.05, ** *p* ≤ 0.01, *** *p* ≤ 0.001). To keep the graph clear, only particular significances are displayed. (**b**) Microarray was performed according to the manufacturer’s instructions (Alere Technologies GmbH [now Abbott Rapid Diagnostics GmbH, Jena, Germany]). While dark gray squares indicate genes that were detected, white squares mark negative results. Light gray squares represent genes whose presence was not determined. Abbreviations: ANOVA (analysis of variance), *aur* (aureolysin), CFU (colony forming units), *edinA/B/C* (epidermal cell differentiation inhibitor A/B/C), *egc* (enterotoxin gene cluster), *etA/B/D* (exfoliative toxin A/B/D), *lukD/E/FM/S/X/Y* (leukocidin D/E/F/M/S/X/Y), *lukF/S-PV* (Panton-Valentine leukocidin F/S), HA-MRSA (hospital-associated methicillin-resistant *S. aureus*), LA-MRSA (livestock-associated methicillin-resistant *S. aureus*), LDH (lactate dehydrogenase), NC (negative control), *nuc1* (thermonuclease), MSSA (methicillin-susceptible *S. aureus*), S (isolated from sputum), *saeS* (histidine protein kinase of two-component regulatory system *saeR/S*), *sarA* (staphylococcal accessory regulator A), *sea/b/c/d/e/g/h//j/k/l/lm/ln/lo/lu/q/r* (staphylococcal enterotoxin a/b/c/d/e/g/h//j/k/l/lm/ln/lo/lu/q/r), *setB1/B2/B3* (staphylococcal exotoxin), SD (standard deviation), *spa* (staphylococcal protein A), *splA/B/E* (serine protease A/B/E), *sspA* (serine protease), *sspB/P* (staphopain B/A), T (isolated from throat), *tst1* (toxic shock syndrome toxin 1).

**Table 1 toxins-12-00360-t001:** Analysis of bacterial hemolysis on Columbia sheep blood agar. Bacteria were streaked onto Columbia sheep blood agar and incubated at 37 °C, and subsequently 4 °C, for 24 h, to determine the activity of bacterial hemolysins. After incubation at 37 °C, in addition to a clear hemolytic zone, some bacterial isolates showed a dark, diffuse zone of hemolysis, which became clear after a 4 °C incubation (Appendix A).

Investigated Bacteria			37 °C Incubation		4 °C Incubation
Group	*spa*-Type	Isolate	Clear Hemolytic Zone	Dark, Diffuse Hemolytic Zone	Clear Hemolytic Zone
LA-MRSA	t034S	Early	+	++++	++++
		Late	+	++++	++++
	t034T	Early	+	++++	++++
		Late	+	++++	++++
	t011	Early	+	++++	++++
		Late	+	++++	++++
HA-MRSA	t003	Early	+	-	++
		Late	++	-	++
	t010	Early	++	-	+++
		Late	++	-	+++
	t548	Early	-	-	-
		Late	-	++++	++++
MSSA	t617	Early	+	-	+
		Late	-	-	+
	t080	Early	++	-	+++
		Late	+	-	++
	t002	Early	++	-	+++
		Late	++	-	+++

Abbreviations: HA-MRSA (hospital-associated methicillin-resistant *S. aureus*), LA-MRSA (livestock-associated methicillin-resistant *S. aureus*), MSSA (methicillin-susceptible *S. aureus*), *spa* (staphylococcal protein A), S (isolated from sputum), T (isolated from throat). Symbols: - not detectable, + slight intensity, ++ strong intensity, +++ stronger intensity, ++++ very strong intensity.

**Table 2 toxins-12-00360-t002:** Subset of resistance genes detected in LA-MRSA, HA-MRSA, and MSSA CF isolates by microarray analysis. Microarray was performed according to the manufacturer’s instructions (Alere Technologies GmbH [now Abbott Rapid Diagnostics GmbH, Jena, Germany]).

	LA-MRSA						HA-MRSA						MSSA					
*spa*-type	t034S		t034T		t011		t003		t010		t548		t617		t080		t002	
Isolate	Early	Late	Early	Late	Early	Late	Early	Late	Early	Late	Early	Late	Early	Late	Early	Late	Early	Late
Gene																		
*aacA-aphD*																		
*aadD*																		
*aphA3*																		
*blaI/R/Z*																		
*cat*																		
*cfr*																		
*dfrS1*																		
*ermA*																		
*ermB*																		
*ermC*																		
*fexA*																		
*fosB*																		
*inuA*																		
*mecA*																		
*mefA*																		
*mphC*																		
*msrA*																		
*mupA*																		
*tetK*																		
*tetM*																		
*vanA/B/Z*																		
*vatA/B*																		
*vgaA*																		

Abbreviations: *aacA-aphD* (bifunctional 6’-aminoglycoside N-acetyltransferase AAC(6′)-2″-aminoglycoside phosphotransferase APH(2″); resistance to gentamicin, tobramycin, and kanamycin), *aadD* (aminoglycoside adenyltransferase; resistance to kanamycin), *aphA3* (3’-aminoglycoside phosphotransferase type III; resistance to kanamycin), *blaI/R/Z* (β-lactamase repressor, regulator, and gene; resistance to penicillin), *cat* (chloramphenicol acetyltransferase; resistance to chloramphenicol), *cfr* (rRNA methyltransferase; resistance to linezolid, chloramphenicol, florfenicol, and clindamycin), *dfrS1* (resistance to trimethoprim), *ermA/B/C* (rRNA adenine N(6)-methyltransferases; resistance to erythromycin and clindamycin), *fexA* (chloramphenicol exporter; resistance to florfenicol and chloramphenicol), *fosB* (metallothiol transferase; resistance to fosfomycin), HA-MRSA (hospital-associated methicillin-resistant *S. aureus*), *inuA* (resistance to lincosamides), LA-MRSA (livestock-associated methicillin-resistant *S. aureus*), *mecA* (penicillin binding protein 2a; resistance to β-lactam antibiotics, like methicillin), *mefA* (macrolide efflux protein A; resistance to macrolides), *mphC* (macrolide phosphotransferase; resistance to macrolides), *msrA* (ribosomal protection protein; resistance to erythromycin and streptogramin), MSSA (methicillin-susceptible *S. aureus*), *mupA* (isoleucyl-tRNA synthetase; resistance to mupirocin), S (isolated from sputum), T (isolated from throat), *tetK/M* (*tetK*: tetracycline efflux protein/*tetM*: ribosomal protection protein; resistance to tetracycline), *vanA/B/Z* (glycopeptide resistance proteins; resistance to vancomycin), *vatA/B* (acetyltransferases; resistance to streptogramins), *vgaA* (ribosomal protection protein; resistance to streptogramins). Dark gray squares: positive, light gray squares: not determined, white squares: negative.

**Table 3 toxins-12-00360-t003:** Early and late *S. aureus* CF isolates analyzed in the study.

Strain	Patient	Isolate	MLST	*spa*-type	*spa*-type Repeats	Origin	Year of Isolation
LA-MRSA	1	Early	CC398	t034	08-16-02-25-02-25-34-24-25	Sputum	2014
		Late					2017
	2	Early	CC398	t034	08-16-02-25-02-25-34-24-25	Throat	2002
		Late					2010
	3	Early	CC398	t011	08-16-02-25-34-24-25	Sputum	2012
		Late					2016
HA-MRSA	4	Early	CC5	t003	26-17-20-17-12-17-17-16	Sputum	2010
		Late					2016
	5	Early	CC5	t010	26-17-34-17-20-17-12-17-16	Sputum	2013
		Late					2017
	6	Early	CC5	t548	26-23-17-34-17-20-17-12-16	Throat	2008
		Late					2014
MSSA	7	Early	CC30	t617	15-21-16-02-24-24	Throat	2001
		Late				Sputum	2010
	8	Early	CC45	t080	09-02-16-34-42-17-16-34	Throat	1995
		Late					2008
	9	Early	CC5	t002	26-23-17-34-17-20-17-12-17-16	Sputum	1994
		Late					1997

Abbreviations: CC (clonal complex), HA-MRSA (hospital-associated methicillin-resistant *S. aureus*), LA-MRSA (livestock-associated methicillin-resistant *S. aureus*), MLST (multilocus sequence type), MSSA (methicillin-susceptible *S. aureus*), S (isolated from sputum), *spa* (staphylococcal protein A), T (isolated from throat).

**Table 4 toxins-12-00360-t004:** Bacterial laboratory and reference strains used in the study.

Bacterial species	Strain	Application	Reference
*Staphylococcus aureus*	Cowan I	Invasion assay (PC)	ATCC 12598
*Staphylococcus carnosus*	TM300	Biofilm assay (NC)	[46]
*Staphylococcus epidermidis*	RP62A	Biofilm assay (PC)	ATCC 35984

Abbreviations: NC (negative control), PC (positive control).

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
