# Peer review of "The Virulence Potential of Livestock-Associated Methicillin-Resistant Staphylococcus aureus Cultured from the Airways of Cystic Fibrosis Patients"

_toxins, 2020, doi:10.3390/toxins12060360_

Round 1

Reviewer 1 Report

In this manuscript, the authors investigated the virulence potential of livestock-associated methicillin-resistant Staphylococcus aureus (LA-MRSA) within the airways of cystic fibrosis (CF) patients. The overall findings are of great interest and provide new insights into LA-MRSA infection. There are some minor concerns to be addressed. First, the authors analyzed respiratory isolates from CF patients. Pulmonary MRSA infection is currently treated with vancomycin, linezolid or clindamycin. In contrast, tetracycline is used to treated cutaneous MRSA. The authors only provided data on tetracycline resistance of LA-MRSA (Table 2). Could the authors provide additional data on antibiotic sensitivity to clinically irrelevant drugs? This is of importance. Second, did the authors know if tetracycline could be a food additive ingested by livestock sampled in this study? This may help explain the occurrence of tetracycline resistance of LA-MRSA.

Author Response

Comments and Suggestions for Authors: In this manuscript, the authors investigated the virulence potential of livestock-associated methicillin-resistant Staphylococcus aureus (LA-MRSA) within the airways of cystic fibrosis (CF) patients. The overall findings are of great interest and provide new insights into LA-MRSA infection.

Authors' response: We thank the reviewer for the appreciation of our work.

Comments and Suggestions for Authors: There are some minor concerns to be addressed. First, the authors analyzed respiratory isolates from CF patients. Pulmonary MRSA infection is currently treated with vancomycin, linezolid or clindamycin. In contrast, tetracycline is used to treated cutaneous MRSA. The authors only provided data on tetracycline resistance of LA-MRSA (Table 2). Could the authors provide additional data on antibiotic sensitivity to clinically irrelevant drugs? This is of importance.

Authors' response: We agree with the reviewer. More information on antibiotic sensitivity/resistance in the S. aureus isolates would extend the scope of the manuscript. Therefore, in the revised version, we presented more antibiotic resistance genes in Table 2, adapted the Table legend (lines 267-286), added sentences/words in lines 238-243, lines 247-250, lines 339-342 and lines 347-353 and implemented the references 43 (Lekagul et al.,2019) and 44 (Chmiel et al., 2014) to discuss the new data.

Comments and Suggestions for Authors: Second, did the authors know if tetracycline could be a food additive ingested by livestock sampled in this study? This may help explain the occurrence of tetracycline resistance of LA-MRSA.

Authors' response: We thank the reviewer for this suggestion. In the revised version of the manuscript, we addressed this issue and extended the sentence in lines 341-342 and added references 43 (Lekagul et al.,2019). We also included the following sentence:  “Another feature of LA-MRSA is the carriage of the resistance gene tetM, which confers resistance against tetracycline (Price et al., MBio 2012).”

Reviewer 2 Report

Toxins 813122

Introduction - would benefit by a brief discussion of how isolates were identified as “Livestock Associated” (LA-MRSA). WGS? MLST?

Line 36 – replace “nowdays” with “currently”

Line 38 – Remove “Alarmingly, also” and start sentence with Animals from…

Line 61 – remove today and replace with “recently” there is limited

Line 72 - replace “is highly affine to” with “has high affinity for”

Line 86 - strongly

All Figures – the dark blue is so intense it is difficult t discern the striped bars.

Good use of classic microbiology techniques for phenotypic characterizations

Figure 2 suggests that genes other than the ica operon are required. Additional discussion of alternate invasion mechanisms in the text would be helpful. clp?

Author Response

Comments and Suggestions for Authors: Introduction - would benefit by a brief discussion of how isolates were identified as “Livestock Associated” (LA-MRSA). WGS? MLST?

Authors' response: We thank the reviewer for this comment and added a sentence in lines 42-46”.

Comments and Suggestions for Authors:

Line 36 – replace “nowdays” with “currently”

Line 38 – Remove “Alarmingly, also” and start sentence with Animals from…

Line 61 – remove today and replace with “recently” there is limited

Line 72 - replace “is highly affine to” with “has high affinity for”

Line 86 – strongly

Authors' response: We changed the respective words in lines 37, 39, 64 (previously line 61), 78 (previously line 72) and 92 (previously line 86) according to the suggestions of the reviewer.

Comments and Suggestions for Authors: All Figures – the dark blue is so intense it is difficult to discern the striped bars.

Authors' response: We apologize for this inconvenience. We chose a lighter blue to color the bars referring to HA-MRSA.

Comments and Suggestions for Authors: Good use of classic microbiology techniques for phenotypic characterizations

Authors' response: We thank the reviewer for the appreciation of our work.

Comments and Suggestions for Authors: Figure 2 suggests that genes other than the ica operon are required.

Authors' response: The reviewer is right with his/her suggestion. In the discussion part of the first version of the manuscript, we already mentioned some other genes that could be involved in biofilm formation (first version, lines 283-284). However, in the revised version of the manuscript, we added sentences in lines 316-321 and the reference 37 (Fernández-Barat et al. 2018) to discuss the biofilm formation of the analyzed S. aureus strains in more detail.

Comments and Suggestions for Authors: Additional discussion of alternate invasion mechanisms in the text would be helpful. clp?

Authors' response: We agree with the reviewer. In the revised version of the manuscript, we suggest the investigation of virulence regulators or bacterial stress response genes in order to analyze invasion of the S. aureus CF strains in more detail (lines 332-337 and references 39 (Tuchscherr et al., 2015), 41 (Battistoni et al., 2000) and 42 (Gunaratnam et al., 2019)). In addition, we moved the last sentence of the paragraph dealing with bacterial invasion to lines 327-329.